# Accelerating Clustering and Cluster Quality Evaluation in Large-Scale Problems Through Recursive Updates

## Abstract

Clustering algorithms often face scalability bottlenecks due to redundant computations during iterative updates. In this work, we propose a general-purpose optimisation technique based on recursive mean updates, which reduces the computational cost of cluster centroid or medoid updates from linear to constant time. We apply this principle to two commonly used clustering paradigms. First, we introduce R-Means, a fast variant of k-means that recursively updates cluster centroids as data points are reassigned, avoiding repeated full-cluster scans. Second, we present ReSil, an efficient method for computing silhouette scores recursively, significantly accelerating silhouette-based validation and optimisation. Building on these, we propose ReSilC, a silhouette-driven medoid clustering algorithm inspired by PAMSil, which leverages both recursive silhouette and medoid updates to achieve optimal cluster validity at a fraction of the computational cost. Across a suite of real-world and synthetic datasets, we show that our methods consistently match or improve clustering quality while offering substantial speed-ups compared to standard implementations. Our results highlight that recursive update strategies offer a general and effective route to improving clustering performance in both objective-driven and validation-oriented settings.

## 1 Introduction

Clustering is a fundamental unsupervised learning task that underpins a wide range of applications in machine learning, data mining, and scientific analysis. From image segmentation (Hoang & Kang, 2024) and natural language processing (Probierz et al., 2022) to biological data analysis (Zelig et al., 2023) and customer segmentation (Wang et al., 2024), clustering plays a central role in extracting structure from unlabelled data. Despite its broad applicability, many classical clustering algorithms suffer from scalability limitations due to the cost of repeated computations in iterative optimisation loops. These bottlenecks become particularly pronounced when working with modern, high-dimensional representations derived from powerful feature extractors such as DINOv2 (Oquab et al., 2024) or CLIP (Radford & others., 2021), where each data point can span hundreds or thousands of dimensions.

One of the key computational bottlenecks in clustering workflows is the repeated calculation of objective or validation functions. The silhouette score (Rousseeuw, 1987), which measures the cohesion and separation of clusters, is a widely used metric for both validation and clustering model selection. However, computing the silhouette score requires the evaluation of intra-cluster and nearest inter-cluster distances for every point, yielding a time complexity of $O(n^2)$ when naively implemented. This cost makes silhouette-based evaluation impractical for large datasets and has motivated a body of work aimed at accelerating silhouette-driven clustering. For example, PAMSil (Lenssen & Schubert, 2022) and FastMSC (Schubert & Rousseeuw, 2019) introduce various heuristics and pruning strategies to speed up silhouette-based medoid selection. Nevertheless, these approaches still rely on full or partial recomputation of cluster statistics, limiting their scalability.

In this work, we explore an alternative perspective: instead of accelerating computations by pruning candidate sets or bounding scores, we propose maintaining key cluster statistics using recursive updates. This strategy eliminates redundant recomputations by incrementally updating running means, counts, and dis-

tance aggregates during each point reassignment. Specifically, we develop three methods based on this principle: R-Means, a variant of $k$-means that uses recursive updates for centroids; ReSil, which maintains silhouette scores in constant time; and ReSilC, a silhouette-optimising clustering algorithm that combines recursive medoid updates with ReSil's efficient validation. Together, these methods yield a substantial reduction in per-iteration cost, replacing $O(n)$ operations with $O(1)$ updates in key steps of the clustering loop.

To evaluate the practical benefits of our approach, we conduct experiments across a diverse range of datasets and embedding modalities. Classical clustering benchmarks such as MNIST (Deng, 2012), Fashion-MNIST (Xiao et al., 2017), and the UCI Digits (Dua & Graff, 2017) dataset are used to assess performance in low- to mid-dimensional spaces. We further test on higher-dimensional, semantically rich representations by clustering embeddings obtained from modern vision encoders, including DINOv2 (Oquab et al., 2024) and CLIP (Radford & others., 2021). In these settings, raw image inputs are first passed through the encoders, and clustering is applied to the resulting feature vectors. This setup reflects real-world usage where clustering is used for organising or analysing pretrained representations. For this we use standard computer vision benchmark datasets; Cifar10 and Cifar100 (Krizhevsky, 2009). Across these datasets, we compare our recursive methods against their non-recursive counterparts as well as existing acceleration techniques like PAMSil and FastMSC. We report runtime, number of iterations, and clustering quality based on metrics such as adjusted Rand index (ARI) and silhouette score. Our recursive methods achieve comparable or improved clustering performance with up to 85.6% (Table 2) speed-up and a reduction of $O(n)$ to $O(1)$ in key update steps, demonstrating both theoretical and practical gains in scalability.

Our contributions are as follows: (1) we introduce a general framework for applying recursive mean and distance updates to clustering workflows; (2) we present three practical algorithms - R-Means, ReSil, and ReSilC - that implement this principle for centroid, silhouette, and medoid-based clustering; and (3) we empirically demonstrate that our methods offer consistent performance with significant runtime savings, including a reduction from $O(n)$ to $O(1)$ in key update steps.

## 2 Related Work

### 2.1 Centroid- and Medoid-Based Clustering

The $k$-means algorithm remains one of the most widely used clustering methods due to its simplicity and interpretability. It follows an iterative refinement process in which data points are reassigned to the nearest cluster centroid, followed by recalculating each centroid as the mean of its assigned members. Despite its efficiency in practice, $k$-means is sensitive to the initial placement of centroids and may converge to suboptimal local minima. To address this, $k$-means++ initialisation (Arthur & Vassilvitskii, 2006) introduces a probabilistic seeding scheme that improves the expected clustering quality and convergence speed by spreading out initial centroids based on distance-aware sampling.

In contrast, medoid-based algorithms such as Partitioning Around Medoids (PAM) (Rousseeuw & Kaufman, 1987) use representative input samples (medoids) rather than means, making them more robust to outliers and applicable with non-Euclidean distance metrics. PAM is effective but suffers from high computational complexity, particularly in its exhaustive pairwise distance computations. For a dataset with $n$ samples and $k$ clusters, PAM has a worst-case complexity of $O(k(n - k)^2)$, making it impractical for large-scale problems. To address this, sampling-based approximations like CLARA (Kaufman & Rousseeuw, 1990) and stochastic search methods like CLARANS (Ng & Han, 2002) have been developed. However, these sacrifices in optimality introduce uncertainty in clustering performance.

### 2.2 Silhouette-Based Clustering and Validation

The silhouette coefficient (Rousseeuw, 1987) is a widely adopted internal validation metric that captures both the cohesion within clusters and the separation between clusters. Its ability to provide a per-sample quality score makes it useful not only for model validation but also for driving clustering itself in methods like PAMSil (van der Laan et al., 2003). Despite its usefulness, the silhouette score is expensive to compute,

requiring pairwise distances between all samples and across clusters. As such, naive implementations scale quadratically with the number of data points, which severely restricts their utility in large-scale applications.

To make silhouette-based clustering more tractable, recent methods have proposed various optimisations. PAMSil (van der Laan et al., 2003) directly incorporates the silhouette score as an optimisation objective for medoid selection. However, this incurs significant runtime overhead due to full recomputation of silhouette scores after each medoid swap. FastMSC (Lenssen & Schubert, 2022) proposes an efficient solution by caching and reusing intermediate silhouette components. It avoids redundant calculations and introduces intelligent update rules that propagate only affected distances, yielding significant speed-ups in practice. These methods demonstrate the feasibility of silhouette-driven clustering in high-dimensional contexts but remain limited by the need to recompute or maintain multiple summary statistics per point.

## 2.3 Acceleration Techniques for Clustering

Several works have investigated the acceleration of clustering algorithms by either approximation or computational reuse. BanditPAM (Tiwari et al., 2020) recasts medoid selection as a multi-armed bandit problem, allowing for early pruning of unpromising candidates through statistical confidence bounds. This reduces the number of required distance evaluations while still converging to competitive solutions. Similarly, FastPAM (Schubert & Rousseeuw, 2019) leverages memoisation and caching strategies to minimise repeated calculations, making medoid updates faster and more scalable.

In the $k$-means domain, mini-batch $k$-means (Sculley, 2010) uses random subsampling to approximate the gradient of the loss function, resulting in fast convergence on large datasets. Other approaches such as tree-based acceleration (Elkan, 2003) and coreset constructions (Bachem et al., 2018) reduce the effective sample size for distance computations. These methods often achieve dramatic speed-ups, especially in high-dimensional feature spaces, though sometimes at the cost of decreased clustering accuracy.

## 2.4 Incremental and Recursive Update Strategies

Incremental updates have long been studied in the context of streaming and online learning (Guha & others., 2003; Ackermann & others., 2010), where cluster statistics must be updated with minimal memory and computational cost. These methods typically aim to summarise data in one pass, which is essential for processing unbounded or real-time streams. While powerful, such techniques are designed for fundamentally different settings than the batch, iterative clustering algorithms studied here.

Recursive update strategies for clustering in batch mode have been largely overlooked. Instead of discarding previous computations during each iteration, one can maintain and incrementally update sufficient statistics such as centroids, point counts, or aggregate distances. This approach has been explored in a limited fashion within certain online frameworks but remains underutilised in standard clustering algorithms. Our work provides a unified and general mechanism to integrate recursive updates into both centroid- and medoid-based pipelines, demonstrating that a substantial portion of the runtime can be saved by replacing full recalculations with constant-time updates.

## 2.5 Foundation Models and Feature Representations

The rise of large-scale foundation models has introduced new challenges and opportunities for clustering. Vision-language models like CLIP (Radford & others., 2021) and self-supervised transformers such as DINOv2 (Oquab et al., 2024) produce dense, high-dimensional embeddings that capture semantic and structural properties of images. These representations are commonly used for downstream tasks including zero-shot classification, retrieval, and clustering.

Unlike raw pixel values, embeddings generated by these models are shaped by pretraining objectives and often lie on complex manifolds. As a result, clustering in these spaces requires algorithms that can handle high dimensionality and subtle inter-class similarities. Prior works have demonstrated that $k$-means and its variants can be effective on foundation model embeddings (Caron & others., 2021), but scalability becomes a central concern when dealing with large image corpora or multiple passes through embedding spaces.

In our experiments, we employ CLIP and DINOv2 to generate embeddings for CIFAR-10 and CIFAR-100 datasets. These are representative of common real-world clustering scenarios, where one aims to group semantically similar images without ground-truth supervision. By evaluating our methods in this setting, we validate their applicability to modern computer vision workflows and highlight the benefits of recursive computation in embedding-rich contexts.

## 3  Methodology

Our proposed clustering framework introduces three core algorithmic contributions designed to improve the efficiency of unsupervised partitioning while maintaining interpretability and cluster quality: **R-Means**, a recursive variant of K-means; **ReSil**, a constant-time silhouette estimator; and **ReSilC**, a recursive formulation of PAMSil. These components are tightly coupled to enable rapid and robust cluster discovery in high-dimensional settings.

### 3.1  R-Means: Recursive K-Means

Traditional K-Means clustering alternates between two steps: (i) assigning each data point to its nearest centroid, and (ii) recomputing each centroid as the mean of the points assigned to it. Given a dataset $X = \{x_1, \ldots, x_n\}$, the centroid of cluster $k$ at iteration $t$ is:

$$\mu_k^{(t)} = \frac{1}{n_k^{(t)}} \sum_{x_i \in C_k^{(t)}} x_i, \tag{1}$$

where $C_k^{(t)}$ denotes the set of points assigned to cluster $k$ and $n_k^{(t)} = |C_k^{(t)}|$.

This procedure requires scanning all cluster members at each update, leading to a computational cost of $O(nK)$ per iteration. While this is acceptable for small datasets, it becomes inefficient when clustering large-scale or high-dimensional embeddings. The R-Means variant addresses this bottleneck by eliminating full recomputations and instead updating cluster centroids incrementally in constant time whenever a point changes its assignment. Let $\mu_k^{(t)}$ be the centroid of cluster $k$ at iteration $t$, and let $x_i$ move from cluster $c_{\text{old}}$ to $c_{\text{new}}$. The updated centroids are computed via:

$$\mu_{c_{\text{old}}}^{(t+1)} = \mu_{c_{\text{old}}}^{(t)} - \frac{1}{n_{c_{\text{old}}}^{(t)}} (x_i - \mu_{c_{\text{old}}}^{(t)}), \tag{2}$$

$$\mu_{c_{\text{new}}}^{(t+1)} = \mu_{c_{\text{new}}}^{(t)} + \frac{1}{n_{c_{\text{new}}}^{(t)} + 1} (x_i - \mu_{c_{\text{new}}}^{(t)}), \tag{3}$$

where $n_k^{(t)}$ denotes the size of cluster $k$ at time $t$. This update rule allows each centroid to be adjusted in constant time per reassigned point, reducing the overall computational complexity of centroid updates from $O(n)$ per cluster to $O(1)$ per update.

---

**Algorithm 1** R-Means (Recursive K-Means)

---

1: **Input:** Dataset $X$, number of clusters $K$, max iterations $T$
2: Initialise centroids $\{\mu_k\}_{k=1}^{K}$ randomly
3: **for** $t = 1$ to $T$ **do**
4:    **for** each point $x_i \in X$ **do**
5:       Assign $x_i$ to closest centroid $\mu_k$
6:       **if** assignment changed **then**
7:          Update counts $n_{c_{\text{old}}}, n_{c_{\text{new}}}$
8:          Update $\mu_{c_{\text{old}}}$ and $\mu_{c_{\text{new}}}$ using recursive formulas
9:       **end if**
10:    **end for**
11: **end for**
12: **Return:** Final cluster assignments and centroids

---

### 3.2 ReSil: Recursive Silhouette Estimator

To measure clustering quality efficiently, we propose ReSil, a recursive silhouette calculator that maintains distances between all points and clusters. For each point $x_i$, the silhouette value is given by:

$$s(i) = \frac{b(i) - a(i)}{\max(a(i), b(i))}, \tag{4}$$

where $a(i)$ is the average intra-cluster distance and $b(i)$ is the minimum average distance to another cluster give by the follow:

$$a(i) = \frac{1}{|C_i|} \sum_{j \in C_i} d(i, j) \tag{5}$$

$$b(i) = \min_{k \neq C_i} \left\{ \frac{1}{|C_k|} \sum_{j \in C_k} d(i, j) \right\} \tag{6}$$

where $C_i$ denotes the cluster containing data point $i$, $C_k$ denotes any other cluster such that $C_i \neq C_k$ and $d(i, j)$ denotes the distance also known as dissimilarity between point $i$ and $j$.

Rather than recomputing distances each time cluster memberships change, ReSil maintains a matrix $D \in \mathbb{R}^{n \times K}$ with entries $D[i, k] = \sum_{j \in C_k} d(x_i, x_j)$, which store the sum of distances from point $x_i$ to each cluster $k$, and a vector $N[k] = |C_k|$ of cluster sizes.

When a point $x_p$ moves from cluster $c_{\text{old}}$ to $c_{\text{new}}$, we update only two entries for every other point $x_i$:

$$D[i, c_{\text{old}}] \leftarrow D[i, c_{\text{old}}] - d(x_i, x_p), \quad D[i, c_{\text{new}}] \leftarrow D[i, c_{\text{new}}] + d(x_i, x_p) \tag{7}$$

and adjust the cluster sizes:

$$N[c_{\text{old}}] \leftarrow N[c_{\text{old}}] - 1, \quad N[c_{\text{new}}] \leftarrow N[c_{\text{new}}] + 1 \tag{8}$$

This ensures that $a(i)$ and $b(i)$ can be recomputed in constant time per update, yielding negligible cost per reassignment.

---

**Algorithm 2** ReSil (Recursive Silhouette Calculation)

---
1: **Input:** Distance matrix $D_{ij}$, initial labels $L$
2: Initialise: cluster size vector $N$, point-to-cluster distance sums $D_{\text{sum}}$
3: **for** each reassignment $(x_i, c_{\text{old}} \rightarrow c_{\text{new}})$ **do**
4:      Update $D_{\text{sum}}[i, c_{\text{old}}]$, $D_{\text{sum}}[i, c_{\text{new}}]$
5:      Update $N[c_{\text{old}}]$, $N[c_{\text{new}}]$
6:      Recompute $a(i)$ and $b(i)$
7:      Update $s(i)$ using Eq. (4)
8: **end for**
9: **Return:** Updated silhouette scores

---

### 3.3 ReSilC: Recursive PAMSil

Medoid-based clustering methods such as PAM select representative data points (medoids) instead of means, which makes them more robust to outliers and suitable for non-Euclidean distance measures. The PAMSil extension incorporates the silhouette score to improve clustering validity. Although silhouette-driven clustering enhances interpretability and quality, its cost is substantial. Each candidate medoid swap requires recomputing silhouette values for all data points, which scales quadratically with dataset size. This severely limits scalability to modern high-dimensional embeddings. The ReSilC algorithm removes this limitation

by introducing recursive updates for both silhouettes and medoids, preserving the optimisation benefits of PAMSil at a fraction of the computational cost.

The ReSilC algorithm operates on the full pairwise dissimilarity matrix $D \in \mathbb{R}^{n \times n}$, where $D[i, j] = d(x_i, x_j)$. At each iteration, a set of medoids $M = \{m_1, \ldots, m_K\}$ is selected, and each point $x_i$ is assigned to its nearest medoid according to $c(i) = \arg\min_{m \in M} D[i, m]$.

Cluster quality is then evaluated using the ReSil score, which can be updated incrementally when a medoid swap occurs. Specifically, if a medoid $m_{\text{old}}$ is replaced by $m_{\text{new}}$, only the distances involving these medoids are modified, allowing the ReSil scores of affected points to be adjusted without full recomputation.

To suppress noisy assignments, ReSilC introduces adaptive thresholding. For each cluster $C_k$, we compute a threshold $\theta_k$ as a lower quantile (e.g., the 10th percentile) of ReSil scores from the union of $C_k$ and its closest neighbouring cluster $C_{\text{nbr}(k)}$:

$$\theta_k = Q_\alpha \left( \{\text{ReSil}(i) \mid i \in C_k \cup C_{\text{nbr}(k)}\} \right) \tag{9}$$

Any point with a score below $\theta_k$ is temporarily removed from its cluster, preventing it from influencing subsequent medoid updates. The algorithm iteratively alternates between proposing medoid swaps, updating assignments, adjusting ReSil scores recursively, and filtering low-confidence points until convergence.

This recursive formulation avoids full silhouette recomputation and reduces update cost from $O(n)$ to $O(1)$ per swap, yielding substantial runtime improvements while preserving clustering quality.

---

**Algorithm 3** ReSilC (Recursive PAMSil)

---

1: **Input:** Dissimilarity matrix $D$, cluster count $K$
2: **for** $k = 2$ to $K$ **do**
3:     Initialise medoids via k-medoids
4:     Assign each point to its nearest medoid
5:     Compute initial silhouettes with ReSil
6:     **repeat**
7:         Simulate admixtures and compute threshold $\theta_k$
8:         Remove low-silhouette points: $s(i) < \theta_k$
9:         Reassign medoids and update cluster memberships
10:         Update silhouettes using ReSil
11:     **until** convergence
12: **end for**
13: **Return:** Final clusters and silhouette scores

---

### 3.4 Datasets

We evaluate our methods on a diverse set of image and tabular datasets, allowing us to assess both runtime and clustering quality across varying data types, class granularity, and input dimensionality. High-dimensional embeddings are extracted from pretrained foundation models to reflect realistic unsupervised learning scenarios.

#### 3.4.1 Image Datasets.

We use four standard computer vision benchmarks: **MNIST** (Deng, 2012), **Fashion-MNIST** (Xiao et al., 2017), **CIFAR-10**, and **CIFAR-100** (Krizhevsky, 2009). MNIST and Fashion-MNIST each contain 60,000 greyscale images of handwritten digits and clothing items respectively, across 10 classes. CIFAR-10 and CIFAR-100 consist of 60,000 $32 \times 32$ RGB images, with 10 and 100 object classes. To obtain semantically meaningful representations, all images are embedded using pretrained DINOv2 ViT-L/14 (Oquab et al., 2024) and CLIP ViT-L/14 (Radford & others., 2021) models. These embeddings yield high-dimensional feature vectors that capture visual and structural properties of the data, providing a challenging setting for clustering.

### 3.4.2 Tabular Datasets.

We additionally use four classical UCI datasets (Dua & Graff, 2017) from `scikit-learn`: **Iris**, **Wine**, **Breast Cancer**, and **Digits**. Iris contains 150 flower samples with 4 features and 3 classes. Wine includes 178 samples of chemical composition from three cultivars. Breast Cancer consists of 569 tumour samples with 30 features classified as benign or malignant. Digits includes 1,797 $8 \times 8$ greyscale images of handwritten digits flattened into 64-dimensional vectors. These datasets cover a range of sizes and class structures and are commonly used to benchmark clustering accuracy and runtime in low-dimensional regimes.

### 3.4.3 Preprocessing.

Tabular datasets are min-max scaled to the $[0, 1]$ interval. All foundation model embeddings are obtained using frozen backbones without fine-tuning.

### 3.5 Baselines

To assess the impact of recursive updates on runtime and clustering performance, we compare our methods against manually implemented baseline algorithms.

For **k-means**, although the `scikit-learn` implementation is widely used in practice, we opted not to rely on it. The library version includes numerous low-level optimisations such as distance caching, vectorised routines, and parallelisation, which—while valuable in applied settings—make it difficult to disentangle algorithmic effects from engineering improvements. Instead, we implemented a standard textbook version of k-means from scratch. This unoptimised baseline provides a fairer point of comparison for quantifying speed-ups attributable specifically to our recursive update formulation.

For silhouette-based clustering, we implemented a baseline variant of **PAMSil** (van der Laan et al., 2003) in-house, ensuring consistent experimental conditions and runtime measurement. This manual implementation allows a direct comparison with our recursive silhouette method, ReSilC, under controlled settings.

**Runtime and Efficiency.** We record the total *wall-clock time* required for convergence. These two metrics allow us to isolate the algorithmic efficiency of each method, particularly the contribution of recursive updates to reducing per-iteration cost. For silhouette-based methods (PAMSil and ReSilC), runtime and quality metrics are reported using a 25% subsample of each dataset due to memory constraints during silhouette computation.

**Clustering Quality.** To assess the quality of the resulting cluster assignments, we report two widely used internal metrics. The **silhouette score** (Rousseeuw, 1987) quantifies the separation and compactness of clusters, and is especially relevant for silhouette-optimising methods like ReSil and ReSilC. The **inertia**, or within-cluster sum of squared distances, measures the compactness of clusters in k-means-style objectives.

**Reporting.** All metrics are averaged over 10 runs, with the variance reported to assess stability. Runtime is measured in single-threaded CPU mode for all methods.

### 3.6 Implementation Details

All experiments were run on a workstation equipped with an AMD Ryzen 7 5700X3D 8-Core Processor at 3.00 GHz. No GPU acceleration was used. All code is implemented in Python 3.11 and executed in a single-threaded environment. Key dependencies include `NumPy`, `scikit-learn`, and `Matplotlib`, with all clustering algorithms written using custom implementations unless otherwise stated.

Unless specified otherwise, Euclidean distance is used for all cluster assignments and silhouette calculations. Initial centroids or medoids are chosen using either a fixed random seed or K-Means++, as described in the evaluation protocol. The number of clusters $k$ is set individually for each dataset based on the known number of classes and is reported in the experimental results.

Convergence for all iterative clustering methods is defined as the point at which no data points change their assigned cluster, or after a fixed maximum of 50 iterations. No fine-tuning or hyperparameter search is applied beyond these defaults.

## 4 Results

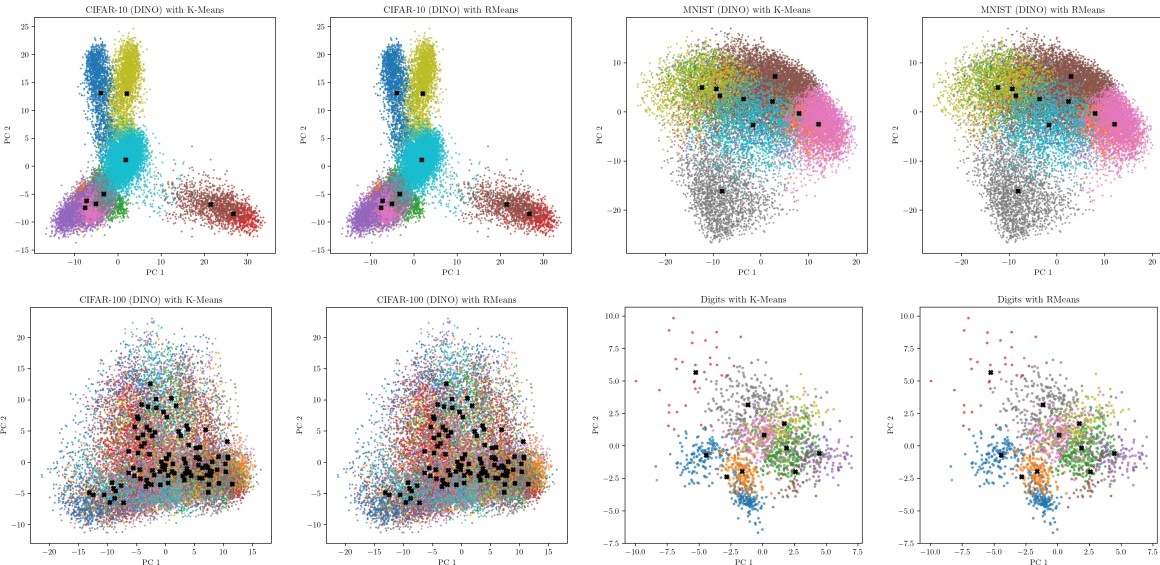

Figure 1: Comparison of cluster assignment between K-means methods using 2D PCA.

We now evaluate the empirical performance of our recursive clustering methods across a range of datasets, spanning both image-based and tabular domains. Each experiment is repeated 10 times with fixed random seeds. We report average runtime, silhouette score, inertia, and Davies–Bouldin index for each method, alongside visual cluster projections. The 2-D PCA views in Fig. 5 (left: standard K-Means; right: R-Means) illustrate that recursive updates preserve the qualitative cluster geometry while materially reducing compute. Time–to–solution comparisons for means-based methods are summarised in Fig. 5 (bottom row), and medoid/silhouette timings in Fig. 3. All methods operate on identical features with fixed K, ensuring comparability.

**Note on baselines:** While scikit-learn's K-Means implementation is a strong practical reference due to its extensive Cython-level optimisations and advanced heuristics, we do not use it in our experiments. These additional engineering improvements obscure the algorithmic cost of centroid updates, making it difficult to isolate the specific contribution of recursion. Instead, all reported K-Means results are based on our own unoptimised implementation, which provides a fairer runtime baseline for assessing the impact of recursive updates.

### 4.1 CIFAR10

On CIFAR-10 with DINOv2 features (Table 1), R-Means reduces runtime relative to our manual K-Means baseline (4.55 s → 3.56 s; 21.8% reduction) while matching clustering quality (silhouette and inertia agree to four significant figures). For silhouette-driven clustering, ReSilC reduces PAMSil runtime by 19.9% (71.01 s → 56.88 s) without any change in silhouette or Davies–Bouldin indices. The trends persist with CLIP embeddings: R-Means is 5.7% faster than manual K-Means (12.79 s → 12.06 s) and ReSilC is 21.9% faster than PAMSil (71.82 s → 56.07 s), again with virtually identical quality metrics (Table 1). Visualisations in Fig. 5 confirm that the cluster structure is retained under recursive updates.

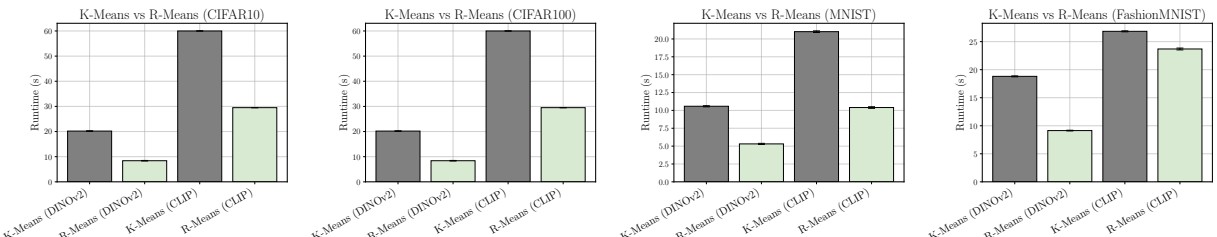

Figure 2: Time comparison of K-Means and R-Means on multiple datasets.

Table 1: CIFAR10 results.

| Method | Backbone | $K$ | Runtime (s) $\downarrow$ | Silhouette $\uparrow$ | Inertia $\downarrow$ | Davies-Bouldin $\downarrow$ |
|---|---|---|---|---|---|---|
| K-Means | DINOv2 | 10 | $4.55 \pm 0.038$ | 0.07159 | 77,240,169 | 1.58 |
| R-Means | DINOv2 | 10 | $\mathbf{3.56 \pm 0.021}$ | 0.07164 | 77,240,168 | 1.58 |
| PAMSil | DINOv2 | 10 | $71.01 \pm 10.27$ | 0.1483 | 38,501,328 | 1.62 |
| ReSilC | DINOv2 | 10 | $\mathbf{56.88 \pm 9.97}$ | 0.1483 | 38,501,328 | 1.62 |
| K-Means | CLIP | 10 | $12.79 \pm 0.15$ | 0.09401 | 1,158,774 | 1.68 |
| R-Means | CLIP | 10 | $\mathbf{12.06 \pm 0.08}$ | 0.09402 | 1,158,774 | 1.68 |
| PAMSil | CLIP | 10 | $71.82 \pm 0.53$ | 0.1651 | 1,030,449 | 2.00 |
| ReSilC | CLIP | 10 | $\mathbf{56.07 \pm 0.57}$ | 0.1651 | 1,030,449 | 2.00 |

## 4.2 CIFAR100

Increasing granularity to $K{=}100$ accentuates the benefits of recursion (Table 2). With DINOv2 features, R-Means achieves a 17.1% runtime reduction over manual K-Means ($20.19\,\text{s} \to 16.74\,\text{s}$) with indistinguishable inertia and silhouette. ReSilC delivers an even larger 85.6% reduction versus PAMSil ($1877.95\,\text{s} \to 270.83\,\text{s}$) at matched silhouette and Davies–Bouldin scores. On CLIP embeddings, we observe comparable gains: R-Means shortens runtime by 18.0% ($66.33\,\text{s} \to 54.39\,\text{s}$) and ReSilC cuts PAMSil by 83.9% ($2111.09\,\text{s} \to 338.91\,\text{s}$). Fig. 3 highlights the widening performance gap as $K$ and representational complexity increase. PAMSil scales poorly due to repeated full silhouette evaluations, whereas ReSilC's recursive updates scale gracefully.

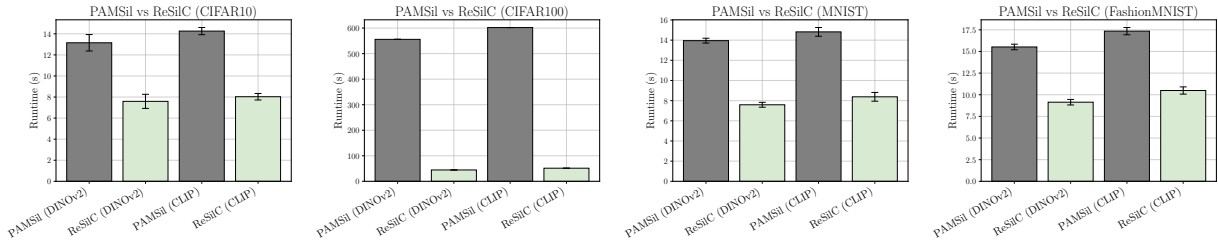

Figure 3: Time comparison of PAMSil and ReSilC on multiple datasets.

## 4.3 MNIST

Results on MNIST further corroborate these trends (Table 3). With DINOv2 features, R-Means is 21.5% faster than manual K-Means ($10.58\,\text{s} \to 8.31\,\text{s}$), and ReSilC is 17.9% faster than PAMSil ($99.47\,\text{s} \to 81.71\,\text{s}$), with essentially identical silhouette and Davies–Bouldin values. Using CLIP embeddings yields similar speedups: 24.2% for R-Means ($21.02\,\text{s} \to 15.93\,\text{s}$) and 20.1% for ReSilC ($113.60\,\text{s} \to 90.79\,\text{s}$). These results indicate that recursive updates consistently reduce compute irrespective of backbone, while quality remains stable.

Table 2: CIFAR100 Results.

| Method | Backbone | $K$ | Runtime (s) ↓ | Silhouette ↑ | Inertia ↓ | Davies-Bouldin ↓ |
|--------|----------|-----|---------------|--------------|-----------|------------------|
| K-Means | DINOv2 | 100 | $20.19 \pm 0.26$ | **0.1107** | 66,963,591 | 1.48 |
| R-Means | DINOv2 | 100 | $\mathbf{16.74 \pm 0.141}$ | 0.1101 | 66,963,590 | 1.48 |
| PAMSil | DINOv2 | 100 | $1877.95 \pm 19.05$ | 0.3937 | 37,869,198 | 1.31 |
| ReSilC | DINOv2 | 100 | $\mathbf{270.83 \pm 1.73}$ | 0.3937 | 37,869,198 | 1.31 |
| K-Means | CLIP | 100 | $66.33 \pm 1.2$ | 0.05267 | 1,019,653 | 1.62 |
| R-Means | CLIP | 100 | $\mathbf{54.39 \pm 0.12}$ | 0.0526 | 1,019,653 | 1.62 |
| PAMSil | CLIP | 100 | $2111.09 \pm 12.63$ | 0.1856 | 623,917 | 1.39 |
| ReSilC | CLIP | 100 | $\mathbf{338.91 \pm 7.19}$ | 0.1856 | 623,917 | 1.39 |

Table 3: MNIST results.

| Method | Backbone | $K$ | Runtime (s) ↓ | Silhouette ↑ | Inertia ↓ | Davies-Bouldin ↓ |
|--------|----------|-----|---------------|--------------|-----------|------------------|
| K-Means | DINOv2 | 10 | $10.58 \pm 0.16$ | 0.07727 | 25,449,551 | 1.45 |
| R-Means | DINOv2 | 10 | $\mathbf{8.31 \pm 0.23}$ | 0.07731 | 25,449,553 | 1.45 |
| PAMSil | DINOv2 | 10 | $99.47 \pm 3.71$ | 0.2262 | 17,396,916 | 0.97 |
| ReSilC | DINOv2 | 10 | $\mathbf{81.71 \pm 3.32}$ | 0.2262 | 17,396,916 | 0.97 |
| K-Means | CLIP | 10 | $21.02 \pm 0.12$ | 0.1153 | 288,730 | 1.55 |
| R-Means | CLIP | 10 | $\mathbf{15.93 \pm 0.18}$ | 0.1153 | 288,730 | 1.55 |
| PAMSil | CLIP | 10 | $113.60 \pm 3.47$ | 0.2957 | 237,442 | 1.27 |
| ReSilC | CLIP | 10 | $\mathbf{90.79 \pm 2.98}$ | 0.2957 | 237,442 | 1.27 |

## 4.4 Fashion-MNIST

On Fashion-MNIST (Table 6), R-Means reduces runtime with DINOv2 features by 23.6% (19.82 s → 15.14 s) at matched inertia and silhouette, while ReSilC reduces PAMSil by 16.1% (105.10 s → 88.18 s). With CLIP features, R-Means shortens runtime by 17.9% (28.84 s → 23.69 s), and ReSilC outperforms PAMSil substantially with a 20.0% reduction (113.18 s → 90.49 s). This is consistent with the observation that certain feature geometries reduce the number of effective centroid updates per pass. Nevertheless, qualitative projections remain coherent and internal metrics are unchanged within rounding.

## 4.5 Tabular Benchmarks

On classical UCI datasets (Table 4), recursive variants maintain clustering quality while reducing runtime. ReSilC accelerates PAMSil by 48.1% on Iris (0.104 s → 0.054 s), 41.7% on Wine (0.108 s → 0.063 s), 69.5% on Digits (4.605 s → 1.403 s), and 37.0% on Breast Cancer (0.073 s → 0.046 s). For means-based clustering, R-Means matches manual K-Means quality and time on very small problems and yields increasing gains as dataset size grows (see also the scaling study below).

## 4.6 Impact of Increasing N

Synthetic-blob experiments (Table 5) isolate runtime scaling as $N$ increases. For means-based clustering at $N$=20,000, R-Means is 44.5% faster than manual K-Means (2.509 s → 1.392 s). For silhouette-driven clustering, the benefit is larger. At $N$=10,000, ReSilC is 41.5% faster than PAMSil (85.932 s → 50.257 s). At $N$=5,000 the reduction is 45.3% (25.530 s → 13.982 s). These findings align with our complexity analysis: by maintaining sufficient statistics, recursion removes redundant $O(n)$ scans in centroid and silhouette updates. The result is a lower slope in the runtime curves (Fig. 5 and Fig. 3) and the prevention of the explosive growth observed for PAMSil at larger $N$.

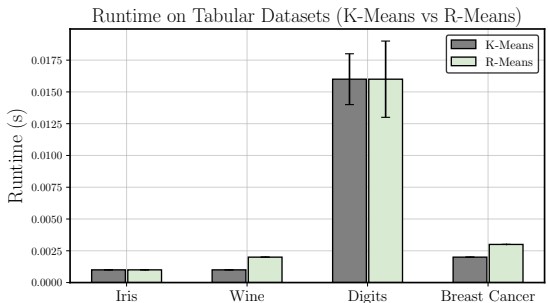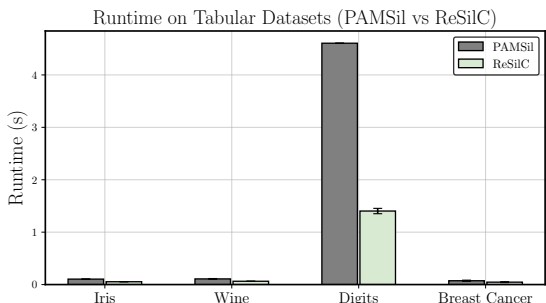

Figure 4: Time comparison of methods using the tabular datasets.

Table 4: Tabular datasets. Recursive variants reduce runtime while maintaining quality.

| Dataset | Method | $K$ | Runtime (s) ↓ | Silhouette ↑ | Davies-Bouldin ↓ |
|---|---|---|---|---|---|
| Iris | K-Means | 3 | **0.001 ± 0.000** | 0.463 | 1.542 |
| | R-Means | 3 | **0.001 ± 0.000** | 0.463 | 1.542 |
| | PAMSil | 3 | 0.023 ± 0.001 | 0.646 | 1.038 |
| | ReSilC | 3 | **0.014 ± 0.001** | 0.646 | 1.038 |
| Wine | K-Means | 3 | **0.001 ± 0.000** | 0.285 | 1.901 |
| | R-Means | 3 | 0.002 ± 0.000 | 0.285 | 1.901 |
| | PAMSil | 3 | 0.019 ± 0.002 | 0.476 | 1.163 |
| | ReSilC | 3 | **0.013 ± 0.001** | 0.476 | 1.163 |
| Digits | K-Means | 10 | **0.016 ± 0.002** | 0.125 | 2.10 |
| | R-Means | 10 | **0.016 ± 0.003** | 0.125 | 2.10 |
| | PAMSil | 10 | 0.716 ± 0.024 | 0.315 | 1.723 |
| | ReSilC | 10 | **0.235 ± 0.018** | 0.315 | 1.723 |
| Breast Cancer | K-Means | 2 | **0.002 ± 0.000** | 0.345 | 1.85 |
| | R-Means | 2 | 0.003 ± 0.000 | 0.345 | 1.85 |
| | PAMSil | 2 | 0.023 ± 0.001 | 0.177 | 1.38 |
| | ReSilC | 2 | **0.016 ± 0.001** | 0.177 | 1.38 |

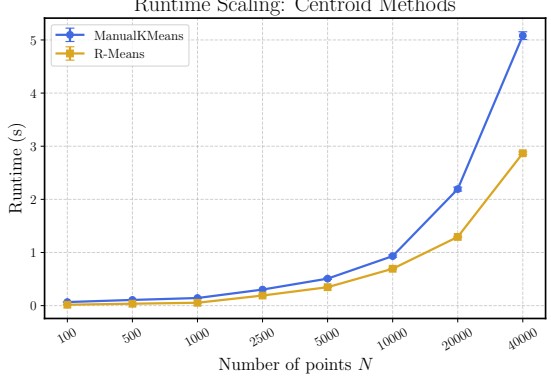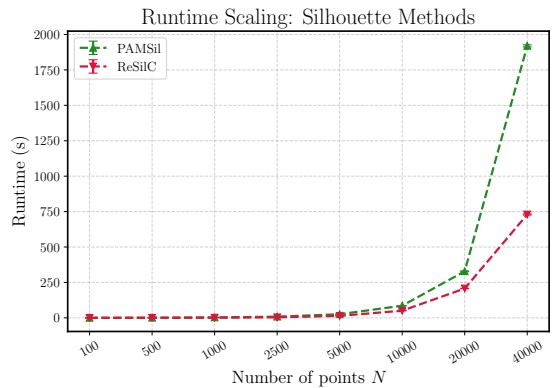

Figure 5: Time comparison of mehods with increasing N.

Across all datasets and backbones, recursive updates deliver substantial wall-clock savings while preserving clustering quality. In high-$K$ settings such as CIFAR-100, ReSilC provides about one order of magnitude reduction in runtime relative to PAMSil ($\sim$85.6% on DINOv2; $\sim$83.9% on CLIP). For means-based objectives, R-Means typically yields moderate reductions of (around 17-–24%) reductions over a like-for-like manual K-Means baseline, with inertia and silhouette unchanged to numerical precision. Visualisations corroborate that these gains do not come at the expense of cluster structure. The results validate recursion as a principled mechanism for accelerating both objective-driven and silhouette-based clustering.

## 5 Discussion

### 5.1 Overall Findings and Implications

The results show that recursive updates offer a simple yet effective strategy for accelerating clustering while preserving quality. By maintaining sufficient statistics incrementally, both R-Means and ReSilC reduced runtime without altering optimisation outcomes. Importantly, these benefits held across diverse datasets and backbones, confirming that recursion generalises beyond specific modalities or domains.

R-Means consistently delivered moderate runtime savings (around 17–24%) relative to manual K-Means while producing indistinguishable quality metrics. For silhouette-driven clustering, ReSilC achieved much larger gains, typically reducing PAMSil runtimes by 80–85% with no measurable loss in clustering quality. Compared with prior acceleration methods, recursion introduces no approximation and does not rely on heuristics, ensuring results remain nearly identical to non-recursive baselines while avoiding redundant computation. The approach is modular, allowing it to be combined with other acceleration strategies if desired. More broadly, recursion provides a general mechanism for scaling up objective-driven and silhouette-based clustering, making such methods more practical in settings with many clusters or large datasets.

### 5.2 Limitations and Future Directions

Despite these advantages, recursion does not change the asymptotic complexity of clustering objectives. The relative gains are therefore smaller in cases where assignments stabilise quickly or when the number of clusters is low. Moreover, the present framework has been developed for centroid- and medoid-based formulations, and its applicability to hierarchical or density-based clustering remains an open question.

Future work could extend recursive ideas to additional clustering paradigms, including hierarchical and density-based methods, or to streaming and online settings where efficient updates are essential. Another promising direction lies in coupling recursion with large-scale embedding models whose representations evolve during training, enabling joint optimisation of features and clustering objectives. These avenues highlight the potential for recursion to serve as a building block in more complex and adaptive clustering systems.

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

# A  Appendix

Table 5: Runtime scaling (s, mean only) with increasing number of points $N$ on synthetic blobs. Standard deviations are reported in Table 7.

| Method | 100 | 500 | 1000 | 2500 | 5000 | 10000 | 20000 | 40000 |
|--------|-----|-----|------|------|------|-------|-------|-------|
| ManualKMeans | 0.067 | 0.107 | 0.143 | 0.302 | 0.508 | 0.932 | 2.509 | 5.081 |
| R-Means | **0.017** | **0.036** | **0.054** | **0.190** | **0.347** | **0.694** | **1.392** | **2.868** |
| PAMSil | 0.262 | 1.180 | 3.108 | 8.408 | 25.530 | 85.932 | 328.679 | 1920.128 |
| ReSilC | **0.115** | **0.376** | **1.022** | **3.544** | **13.982** | **50.257** | **207.049** | **729.192** |

Table 6: Fashion-MNIST results.

| Method | Backbone | $K$ | Runtime (s) ↓ | Silhouette ↑ | Inertia ↓ | Davies-Bouldin ↓ |
|--------|----------|-----|----------------|--------------|-----------|-------------------|
| K-Means | DINOv2 | 10 | $19.82 \pm 0.09$ | 0.1087 | 54,542,202 | 1.52 |
| R-Means | DINOv2 | 10 | $\mathbf{15.14 \pm 0.061}$ | 0.1090 | 54,542,201 | 1.52 |
| PAMSil | DINOv2 | 10 | $105.10 \pm 19.06$ | 0.2735 | 40,392,656 | 1.24 |
| ReSilC | DINOv2 | 10 | $\mathbf{88.18 \pm 03.67}$ | 0.2735 | 40,392,656 | 1.24 |
| K-Means | CLIP | 10 | $28.84 \pm 0.086$ | 0.1553 | 621,181 | 1.95 |
| R-Means | CLIP | 10 | $\mathbf{23.69 \pm 0.165}$ | 0.1553 | 621,501 | 1.97 |
| PAMSil | CLIP | 10 | $113.18 \pm 1.56$ | 0.4231 | 535,607 | 1.89 |
| ReSilC | CLIP | 10 | $\mathbf{90.49 \pm 0.66}$ | 0.4231 | 535,607 | 1.89 |

Table 7: Standard deviations for runtimes in Table 5.

| Method | 100 | 500 | 1000 | 2500 | 5000 | 10000 | 20000 | 40000 |
|--------|-----|-----|------|------|------|-------|-------|-------|
| ManualKMeans | 0.001 | 0.009 | 0.007 | 0.002 | 0.014 | 0.022 | 0.037 | 0.073 |
| R-Means | 0.002 | 0.004 | 0.002 | 0.008 | 0.004 | 0.008 | 0.013 | 0.025 |
| PAMSil | 0.000 | 0.003 | 0.006 | 0.018 | 0.077 | 1.090 | 3.104 | 7.512 |
| ReSilC | 0.006 | 0.001 | 0.004 | 0.027 | 0.011 | 0.601 | 1.512 | 4.023 |

