# OpenReview forum: "Accelerating Clustering and Cluster Quality Evaluation in Large-Scale Problems Through Recursive Updates"
_TMLR — Withdrawn by Authors_

### Review · Reviewer_mkMn · 2025-10-03

**Summary Of Contributions:**

The paper presents recursive algorithm to improve runtime of clustering algorithms like k-means, silhouette score calculation and silhouette based medoid calculation algorithm. The paper provides evaluations on diverse set of datasets and embedding representation s showing improvements in run-time with no drop in clustering performance.

Strengths:
- Presents a simple recursive algorithm to improve the run time of clustering algorithms.
- Presents extensive experimentation with diverse datasets and embedding representations.
- The paper is well written and easy to follow.

Weaknesses:
- The results section first paragraph is a bit confusing with possible mistakes in the referred figure numbers.
- The authors only provide complexity analysis and do not provide a analysis of the storage. It will be interesting to see if the recursive algorithms result in extra storage space requirements, if so by what percentage and if they have a trend with dataset size and embedding representations as well.
- The cluster assignment complexity still remains the same which is a large part of the complexity of the clustering algorithms.

**Audience:**

Yes

**Audience Explanation:**

The paper provides recursive algorithms for improving runtimes of clustering algorithms. Moreover, they show rigorous evaluation on diverse datasets and representations showing the generalizability of the algorithm.Thus the paper contributes theoretically while also showing empirical results of improvements.

**Claims And Evidence:**

Yes

**Claims Explanation:**

The paper provides evaluations on diverse set of datasets and embedding representation s showing improvements in run-time with no drop in clustering performance.

**Requested Changes:**

- Improve clarity of the starting paragraph of the results section.
- Include a discussion regarding space complexity of the recursive algorithms, with percentage increase in space requirements for the recursive algorithms and analysis of its trends with different datasets, clusters, and embedding representations.

---

### Review · Reviewer_kAfc · 2025-10-07

**Summary Of Contributions:**

The authors propose two new algorithm variants, titled R-means and ReSilC.

They claim these run "at a fraction of the computational cost", "from linear to constant time", and "offering substantial speedups" and "a reduction from O(n) to O(1) in key update steps". Unfortunately, these improvements may be a huge overstatement, and in the conclusions the authors eventually admit that "recursion does not change the asymptotic complexity". In fact, the O(n) to O(1) claim may simply come from not realizing they use a vectorized operation that costs O(N) in a single line of code...

The paper has **severe issues**, detailed below. In particular, very well-known previous work (MacQueen 1967, Hartigan and Wong 1979) has not been considered, the methods lack crucial details, do not have a proper complexity analysis despite above claims, and the authors avoided any comparison to existing implementations.

I suggest to revise this paper completely and propose the decision to be **reject and resubmit as new** because of the scope of the necessary changes.

**Additional Comments:**

I suggest to revise this entirely and resubmit as a new article, focusing solely on ReSilC; with a proper discussion of complexity and integration into existing implementations to make this more comparable.

**Audience:**

No

**Audience Explanation:**

If the issues discussed are fixed and studied in more detail, it may become of interest to the community.
But the lack of acknowledging previous work (in particular McQueen, Hartigan and Wong, see below) adds more confusion than contribution.

The name "recursion" is also nonsense. It is a classic incremental computation (of fixed depth), not a recursion (of repeated computation to a base case). Such incremental algorithms are very common in streaming k-means.

There MAY actually be something valuable in ReSilC, but there are too many details missing to derive the actual complexity. My feeling is that it can be a real improvement over PAMSIL, but will not be competitive with FastMSC/FasterMSC. But this needs to be done much more thoroughly.

There appear to be major implementation issues. The authors were not able to apply some of their methods on the 60000 instances data sets, but had to downsample to 25%. A full 60k distance matrix should use about 6.7 GB in triangular form with fp32, and a system with 8 GB hence should still be able to compute Silhouette on such data even with a stored distance matrix...

**Broader Impact Concerns:**

I see no ethical concerns that would require such a statement.

**Claims And Evidence:**

No

**Claims Explanation:**

However, even on their own experiments, the savings are often just about 15%, and clearly not in this order.

The methods in parts lack a proper review of related work, the complexity claims appear to be incorrect.

There may be a bug in the presented algorithm. Others lack detail, such as ReSilC which does not explain how medoids are chosen.

Details are provided below.

**Requested Changes:**

1. Properly relate R-Means to earlier work.
The R-Means method is all but novel. I suggest to drop this from the paper altogether, as it lacks the proclaimed novelty.
While indeed *textbooks* recompute the means every step - and the occasional implementation may do so because of vectorization and/or numerical concerns - it is a *common* technique to only compute the differences from one iteration to the next.
The authors *conjecture* that this "have been largely overlooked".
Unfortunately, it is rather the authors that overlooked a lot of related work. For example, MacQueen,  Some Methods for classification and Analysis of Multivariate Observations, 1967 (the name-giving paper for k-means) already uses this type of computations.
  See for example the source code of kmeans in R: <https://github.com/wch/r-source/blob/3c1d35f627eae8a9ce8dc06fdafa1b28efc3e16f/src/library/stats/src/kmeans.c#L128-L129>
```
cen[iold+k*c] += (cen[iold+k*c] - x[i+n*c])/nc[iold];
cen[inew+k*c] += (x[i+n*c] - cen[inew+k*c])/nc[inew];
```
Hence, the proposed "novel" R-Means is essentially MacQueen's approach reinvented.
Another example can be found, e.g., here in shogun: <https://github.com/shogun-toolbox/shogun/blob/8f01b2b9e4de46a38bf70cdb603db75ebfd4b58b/src/shogun/clustering/KMeans.cpp#L111-L115>

  In fact, nearest-neighbor assignment is not even optimal, which leads to Hartigan and Wong's Algorithm AS 136: A K-Means Clustering Algorithm (1979). It only transfers points to a neighboring cluster if the resulting variance decreases; with a related equation.
Now this does not even seem to be the smartest way of implementing this, because it needs 4 additions and 2 divisions per reassignment and dimension. Many modern implementation will instead keep the sums instead of the means for aggregation (2 additions per reassignment) and perform a single division (per cluster and dimension) at the end of each iteration. In particular the divisions can noticeably increase the run time.

Now there seems to be a good reason why some implementations do not follow this method. First of all, the full recomputation appears to be better for vectorization (in particular on GPUs). Secondly, repeated computation of deltas may cause numerical problems. Imagine we have really large data, these deltas can be very small compared to the mean, and these errors accumulate.
For a similar reason, one should NOT use the "X²-2XY+X²" trick to compute euclidean distances (which the authors use in their implementation), as this has low numerical precision for certain cases compared to (X-Y)².
The paper "Numerically stable parallel computation of (co-)variance" discusses some of these issues.

But because of above example I argue that this is a well-known implementation variation of k-means, and not a novel "R-means" algorithm at all.

2. Compare to other implementations, and use larger data sets.

The authors avoid comparing to other implementations entirely, arguing that this would be misleading, and instead compare to their own "unoptimized baseline" only. This is BAD, a red flag. This allows you to just make the "baseline" arbitrarily bad. You really need to include some good implementation. Instead, you may rather want to integrate your ideas into a framework such as scikit-learn and "inherit" many of their optimizations. This is what will make it much more comparable. And as discussed above, there can also be good argument to do things differently because of performance (such as reducing the amount of divisions, or numerical precision).
Even though there is some truth to the apples-and-oranges argument, you can still include such an implementation and rather discuss these differences. This will give a more realistic impression.
Now because the scikit-learn clustering code is actually pretty poor, I suggest you rather compare to Greg Hamerly's implementations available at https://github.com/ghamerly/fast-kmeans - this is a fairly wide collection of implementations, or Christian Borgelt's implementations available at https://borgelt.net/cluster.html
I also strongly urge to include improved algorithms such as Hamerly's and Elkan's algorithms. In fact these will already involve some of your "novel" incremental computation. It is not acceptable that you entirely ignore them, even with *incorrect* description. On page 3 it is claimed that Elkan, 2003 uses "tree-based acceleration", but instead this method uses clever bounds to skip points during subsequent iterations (which implies that he computes the change in means, and not recomputes means every iteration). This misdescription raises againt he concern that the authors do not know the prior work on k-means well enough.

The tiny "tabular" data sets are clearly not suitable for run-time performance measurements. For scalability, use an actually large data set and subsample, NEVER rely on synthetic blob data which can be extremely misleading (because a trivial sampling method will work very well!) As you go only up to 40000 instances - which is tiny - a random subset of MNIST will be more meaningful than blobs.

3. Prove complexity claims.

If you claim improvements "from linear to constant time", you ought to provide a proof for this.
In fact, the experiments clearly contradict your claim.
And I am convinced that your claims are in fact incorrect. Likely caused by writing this in Python using vectorization and not realizing that "vectorized" operations may easily take O(N) or O(N²) time.

On page 6, it is claimed that the recursive computation "avoids full silhouette recomputation and reduces update cost from O(n) to O(1)".
However, Equation (7) is applied "for every other point $x_i$". So while this equation is indeed O(1), it is performed 2N times for every point reassigned (which is O(N) per iteration). Hence I claim that the complexity of ReSil is O(N²) - just like the naive approach.
Note that the provided pseudocode for ReSil is incorrect and does not agree with the description above. There needs to be an outer loop for every point that is reassigned ($x_p$) and then an inner loop for every other point ($x_i$).

4. Proof correctness of ReSilC.

I have major concerns regarding the correctness of ReSilC. The authors assume that a nearest-neighbor assignment in k-means style will be optimal and yield the same result as PAMSil. This makes sense if you only look at a single point -- minimizing a(i), maximizing b(i). However, in Silhouette every point affects every other point. Moving one point to another cluster to improve its Silhouette may worsen the Silhouette of many other points, hence decreasing the objective. All of this is not thoroughly discussed. Removing R-means from the paper will open some space to discuss this much more thoroughly.
Likely this naive reassignment caused instability, which was compensated for with what the authors call "suppress noisy assignments".
Unfortunately, the paper never explains how the actual cluster medoids are chosen. It seems to be swap-based as in PAMSil. But it is hard to tell the actual complexity if such crucial parts are missing.

5. Compare ReSilC to state-of-the-art

Again the authors avoid comparing to state-of-the-art implementations. They neither consider FasterPAM, BanditPAM, FasterMSC, or DynMSC; all of which are easily available in the Python ecosystem. I can only guess this is because it turned out that these simply work much better. You can always explain differences with implementation details, but do not omit them from the experiments, as these are the true baselines. Some of these have well understood complexities, and comparing the slopes of the methods in a log-log plot is informative even when there is maybe some constant factor difference there due to code optimization.

Furthermore, this comparison begins at a theoretical level. For example FastMSC/FasterMSC as an alternative for Silhouette clustering.
While PAMSil (which according to the FasterMSC paper is even in O(N³k)) indeed recomputes the Silhouette very often, FasterMSC does not seem to do so and claims a runtime of only O(N²). So what is the actual complexity of ReSilC, and how does it experimentally compare to these alternatives? As for the implementation, why not use these implementations as a baseline and add your improvements to them, to make things comparable?

6. Smaller issues

The paper is riddled with small mistakes.
- page 1, "when naively implemented" - by definition, the complexity of Silhouette is O(N²). It is fairly obvious that without additional constraints (such as using metric input) this cannot be improved. Even your approach only handles the computation of a change in Silhouette, not a full recomputation.
- page 1, PAMSil is not by Schubert & Lensen, but van der Laan at al. 2003 (cited correctly on page 2)
- page 1 you complain that other approaches "still rely on full or partial recomputation of cluster statistics". Clearly, so does your approach.
- page 2, "improved clustering performance with up to 85.6%" is that 85.6% of the original run-time, i.e., a saving of about 14%, or is this meant to be a reduction by 85%? But clearly this does not support your claim of "speed-up and a reduction of O(N) to O(1) in key update steps"
- page 2, "implement this principle for centroid, silhouette, and medoid-based clustering", but I do not see k-medoid to be improved here at all.
- page 2, "PAM has a worst-case complexity of O(k(n-k)²)" - per iteration, which could be much worse; and you ignore that this has subsequently been improved to O(n²) per iteration in FastPAM. Why is neither FastPAM nor BanditPAM cited in this discussion of medoid-based clustering, but cited in other places?
- page 3, other methods "remain limited by the need to recompute or maintain multiple summary statistics per point". Reality check, your proposed ReSilC seems to store a N x k matrix with summary statistics per point and medoid... So you do not really overcome this limitation, do you?
- page 3, you now cite BanditPAM and FastPAM, but why do you not compare to them? Nor Elkan (which is not tree-based!), Hamerly, Exponion, Shallot, ... - often NOT at the cost of clustering accuracy.
- page 3-15: several references miss authors. They inconsistently use "et al." or "& others.,". Please name all authors as long as the author list is reasonable in length; have bibtex handle this instead of hard-coding a pseudo-author "others."
- page 4, you claim that clustering CLIP and DINOv2 embeddings is a "common real-word clustering scenario" - but is it really? Can you name actual applications where this has successfully been used, or is this just sounding facing because these are deep-learning?
- page 4, "traditional k-means" - I suggest to use "textbook k-means", as the textbook algorithm is not that commonly used without several modifications.
- page 4, "leading to a computational cost of O(nK) per iteration". This is incorrect. Computing the cluster centers is just O(nd) because every point can only belong to one cluster; it is the cluster reassignment step that needs O(nKd) computations naively.
- page 5, "distance to another cluster give by the follow:" -- check Grammar please.
- page 5, "constant time per update" probably key to your confusion is that you never defined well what an "update" is, nor how many there are. So it may not be as "negligible" as you claim. Measure! What share of the computation time is spent in this step?
- page 6, "at a fraction of the computational cost" - mathematically, 1000/3 is also a fraction... revise to a claim that is supported by both theoretical analysis and empirical evidence.
- page 6, "simulate admixtures" is not defined in the paper.
- page 6, "remove low-silhouette points" - from what, and are they ever added back? Needs to be clearly described to be reproducible
- page 6, "reassign medoids" - not given how the reassignment happens
- page 6, as you loop k=2 to K, I assume you intend to store the best result by Silhouette?
- page 7, tabular data sets -- use much larger data sets.
- page 7, specify clearly which data sets and experiments are subsampled 25% and which are not. Is the resampling properly done only once? We can only infer this from inconsistent run-times on subsequent pages.
- page 7 and following, "averaged over 10 runs, with the variance reported" do NOT report variance, report standard deviation. Judging by the scales of values on the subsequent pages, you indeed give µ+-σ² and not µ+-σ. But it makes NO SENSE to add the variance to the mean.
- page 7, judging from the source code, all these 10 iterations may likely use the same random seed...
- page 7, it was not given how much RAM the system has available, but it is troubling that the authors apparently had to downsample the 60000 images from MNIST to 15000 already. How little RAM did they have to only allow for 15000 instances?
- page 8, "50 iterations" -- it seems scikit-learn defaults to 300 iterations cut-off. Why was this reduced to 50?
- page 8, figure 1 is useless, as one cannot visually detect any differences between the two figures anyway.
- page 8, "materially reducing" - tortured phrase?
- page 8, "Note on baselines": please include them nevertheless in the experiments, as discussed above.
- page 9, In Figure 2 the first two plots clearly are duplicate, and do NOT match the numbers in Table 1.
- page 9, table 1 has extremely low Silhouette values. Clearly, the Silhouette then is not a suitable measure for evaluating these data sets or algorithms - or the algorithms did not actually work. For example, almost every cluster could be a single outlier point. What good are these results when the cluster quality never was asserted?
- page 9, table 1: as written above, the tables list variance, not standard deviation. So 4.58+-0.038 should in fact be 4.58+-0.195.
- page 9, Figure 3 and table 1, it appears that for both PAMSil and ReSilC are run on 25% of the data only. Maybe call these data sets then "CLIP25%" for example.
- page 10, table 2 and 3, this becomes even more pronounced here, as the methods obtain a much smaller inertia than k-means (which optimizes inertia). Clearly, these must be using a 25% subset.
- page 10, "Impact of increasing N": use real data, e.g., MNIST subsets - you go up to 60k only anyway.
- page 10, "prevention of the explosive growth" -- clearly your method in Figure 5 shows the same exponential growth, and does not "prevent" it.
- page 11, Fig 4 and Table 4: it makes little sense to present run-times on the order of 0.001 seconds, as caching and other effects can easily cause larger deviations. For run-times on this scale you need to use careful microbenchmarking (in particular many more repetitions, a warm-up phase, and similar!) -- use larger data sets. Forget iris data!
- page 11, Figure 5: the choice of a log-linear plot here is extremely misleading. Either use a linear-linear, or a log-log plot.
- page 12, the top seems to be a conclusions section, but then it is missing a subheading 4.7 to separate this from the previous subsection.
- page 12, I am not happy that you emphasize your advantage on CIFAR-100, and do not mention that the savings on MNIST are much much smaller.
- page 12, there is a bad math typography in the dashes in "17-–24%"
- page 12-15: replace "and others" with the actual author names. Uppercase MNIST, UCI, PAM, etc. Indicate that Krizhevsky is a TechReport, give the institution and institutional URL instead of a third-party URL such as semanticscholar. For, e.g., Radford, also Tiwari cite the published version instead of the preprint. Add missing DOIs.

Implementation supplementary - did not a full review, but tried to look up some details missing from the paper.
- `self.sum_dist_to_cluster[:, old_c_idx] -= dists_to_idx` -- this is an O(N) vector operation inside a loop `for change_idx, (idx, old_c, new_c) in enumerate(zip(changed_indices, old_labels, new_labels)):` of O(N) iterations -- hence your implementation seems to be O(N²).
- `distances = XX - 2 * np.dot(X, Y.T) + YY` -- this is numerically problematic. Do not use this equation.
- `PAMSIL_SAMPLE_LIMIT = X.shape[0] // 4`, then `if X.shape[0] > PAMSIL_SAMPLE_LIMIT:` -- clearly, you always subsample due to this programming error.

---

### Review · Reviewer_gU1g · 2025-10-10

**Summary Of Contributions:**

The paper proposes recursive update strategies to make three existing methods ($k$-means, silhouette score, PAMSil) more efficient without sacrificing performance.

Strengths:
- simple (in a good way) idea with clear theoretical motivation and practical benefit
- well written
- good breadth of experiments

Weaknesses:
- (possible) mistake or at least ambiguity in 3.2 and 3.3
- no opensource code
- missing some references

**Audience:**

Yes

**Audience Explanation:**

It directly improves the performance of some of the most widely-used clustering methods/metrics and could spur the improvement of others still.

**Claims And Evidence:**

No

**Claims Explanation:**

The claims aren't currently supported well enough, but they're very fixable:

1. Are the authors (reasonably) sure the recursive update in 3.1 isn't already implemented in one of the major toolboxes? I wasn't able to confirm it in the [source code](https://github.com/scikit-learn/scikit-learn/blob/c60dae20604f8b9e585fc18a8fa0e0fb50712179/sklearn/cluster/_k_means_minibatch.pyx), but the [sklearn docs](https://scikit-learn.org/stable/modules/clustering.html#mini-batch-kmeans) for mini-batch $k$-means suggest that it already implements a recursive update:
  > In the second step, the centroids are updated. In contrast to k-means, this is done on a per-sample basis.
2. Don't equations (7) and (8) require 2n operations (so $O(n)$ rather than $O(1)$) per update? "two entries for every other point $x_i$" suggests to me $2(n-1)$ updated entries, and then the two cluster size counts are updated afterwards.
3. In 3.3, it's not clear to me where the savings from $O(n)$ to $O(1)$ comes from, and this section lacks the clear equations comparing non-recursive vs recursive approaches like the last two subsections.

**Requested Changes:**

Critical:
1. Address the three undersupported claims described above.
2. Include a proper [citation of scikit-learn](https://scikit-learn.org/stable/about.html#citing-scikit-learn), and likewise for [numpy](https://numpy.org/citing-numpy/) and [matplotlib](https://matplotlib.org/stable/project/citing.html)

Suggested:
1. provide opensource code reproducing the experiments in the paper and implementing the 3 methods.
2. open a pull request or raise an issue on the scikit-learn repo to get these methods implemented there
3. add some experiments, e.g., using the [big-O package](https://pypi.org/project/big-O/), to corroborate complexity improvements rather in addition to the reported runtimes
4. just before (5): follow -> following
5. in table 2: 0.05267 -> 0.0527

---

### Review · Reviewer_vy9q · 2025-10-13

**Summary Of Contributions:**

This paper proposes a general and practical framework for accelerating iterative clustering algorithms through recursive updates of key statistics such as centroids and distance aggregates. The authors introduce three methods: R-Means, a recursive variant of k-means that updates centroids in constant time upon point reassignment; ReSil, a method for maintaining silhouette scores incrementally; and ReSilC, a recursive version of PAMSil that combines both ideas for efficient, silhouette-optimized medoid clustering.

The claimed core contribution lies in replacing O(n) recomputations with O(1) updates, leading to significant speed-ups, especially in high-dimensional and large-scale settings.

**Audience:**

Yes

**Audience Explanation:**

Accelerating classical clustering algorithms is an important topic and has an active research community. The paper is at least conceptually addressing the concerned problems, yet it needs further justification.

**Claims And Evidence:**

No

**Claims Explanation:**

The effectiveness of all three modules lacks theoretical justification, and the potential side effects of the proposed simplifications, compared to the standard k-means approach, remain unclear.

**Requested Changes:**

1. Computational complexity analyses need to be elaborated, rather than a simple "O(n) -> O(1)" claim.
2. The paper lacks sufficient detail on the implementation of ReSilC, particularly regarding the "adaptive thresholding" and "simulated admixtures" in Algorithm 3. These steps are vague and hinder reproducibility.
3. Please explain where the "recursive" aspect is reflected in the title. All three modules are iterative from my point of view.
4. Please describe how ReSilC selects a set of medoids.
5. The current time comparison may be unfair, as it depends on the number of convergence iterations, which can differ between k-means and the proposed method. Therefore, it would be more informative to report the time cost per fixed number of iterations.

---

### Review · Reviewer_ctx3 · 2025-10-14

**Summary Of Contributions:**

The paper proposes iterative approaches that are claimed to have accelerated K-means, Silhouette calculation, and, applying the previous two, PAMSil algorithms, without sacrificing the quality of the outcome. It's also claimed that the acceleration is not just by a constant, but in terms of time complexity, resolving the often encountered scalability issue of clustering algorithms. Clustering results and runtime on several classic datasets are evaluated and demonstrated in the paper.

Strength
- The motivation of addressing the scalability bottleneck of clustering algorithms is reasonable, and potential impacts are huge.
- The selected datasets are representative.

Weakness
- The effectiveness, or even correctness, of the proposed algorithms are not validated, which is further elaborated with examples in the next section.
- The computation of time complexity is questionable, seemingly taking single step time complexity as overall complexity.
- The introduction of the proposed algorithms are sometimes unclear and not self-contained. For example, for adaptive thresholding, some data points are "temporarily removed from its cluster", but it's unclear when these data points are later considered by the algorithm again, or just discarded ever since.

**Audience:**

No

**Audience Explanation:**

After the issues mentioned above are addressed and the claimed improvement persists, other audience may be interested in the findings. But before that, the conclusions in the paper seem questionable to me.

**Claims And Evidence:**

No

**Claims Explanation:**

1. Several proposed equations introduce errors that are not ignorable in some scenarios, which may even accumulate over iterations.

Take equation (2) in section 3.1 for example, consider a cluster with only two points (-1, 0) and (1, 0) in a 2D space, whose centroid is at (0, 0). Now if (1, 0) leaves the cluster, the centroid should move from (0, 0) to (-1, 0), as (-1, 0) is the only point left in the cluster. However, according to equation (2), the updated centroid is (0, 0) - 0.5 * (1, 0) = (-0.5, 0).

The error in this example is obviously not ignorable, and it's only from a single step, before accumulating through multiple iterations. Similar issues are also observed in several other equations in the paper, e.g. equation (3). Though the errors may be especially severe in specific scenarios, for example when the cluster size is small, some theoretical proof of overall effectiveness or some caveats that the algorithm should come with is needed.

2. The calculation of time complexities are questionable. For example, in section 3.1, it's claimed that "reducing the overall computational complexity of centroid updates from O(n) per cluster to O(1) per update". However, as there could be multiple points changing cluster in a single iteration for the cluster, potentially O(n) with n being the cluster size, the time complexity is at least O(n). Similar problems occur in several other sections, e.g. 3.2, 3.3.

**Requested Changes:**

1. Provide theoretical proof of overall effectiveness of the algorithm, or some caveats that the algorithm should come with.
2. Validate the correctness of the time complexities mentioned in the paper, possibly with each step detailed.
3. Elaborate more clearly about the details of the proposed algorithms. For example, for adaptive thresholding, some data points are "temporarily removed from its cluster", but it's unclear when these data points are later considered by the algorithm again, or just discarded ever since.

---

### Note · Authors · 2025-10-26

**Comment:**

We are withdrawing this submission in order to incorporate the reviewers’ feedback and make substantial revisions. We appreciate the reviewers’ comments and plan to prepare a revised version for future submission.

**Withdrawal Confirmation:**

I have read and agree with the venue's withdrawal policy on behalf of myself and my co-authors.